# Psychological Treatments for Alexithymia: A Systematic Review

**DOI:** 10.3390/bs14121173

**Published:** 2024-12-07

**Authors:** Kanako Tsubaki, Eiji Shimizu

**Affiliations:** Research Center for Child Mental Development, Chiba University, Chiba 260-8670, Japan; eiji@faculty.chiba-u.jp

**Keywords:** alexithymia, treatment, cognitive behavioral therapy, review

## Abstract

Alexithymia, a psychological condition characterized by emotional suppression, is positively correlated with depression and anxiety and can develop into various mental disorders. Although alexithymia affects 10% of the symptomatic population and 25% of psychiatric patients, there has been a paucity of intervention studies. Even though several effective psychological treatments, including cognitive behavioral therapy (CBT), have been tested in recent years, there is a lack of comprehensive reviews on their efficacy. The objectives of this systematic review were to explore and synthesize findings from recent randomized controlled trials (RCTs) about psychological treatments, with the following inclusion criteria: (1) published from 2010 to 2024; (2) full text being available in English; (3) peer-reviewed journals; and (4) baselines and outcomes measured by TAS-20 and raw data were provided. We excluded non-psychological studies and studies involving mindfulness and DBT. We searched electronic databases (PubMed, PsycInfo, and Google Scholar) and found 18 RCTs and 21 arms for alexithymia, with a combined total of 1251 participants. Fourteen arms (67%) investigated the effect of CBT on alexithymia, including acceptance and commitment therapy (seven arms), behavioral activation therapy (two arms), schema therapy, and compassion-focused therapy. The results indicated that most psychological interventions significantly decreased TAS-20, illustrating a showcase of treatments from each trial with different effect sizes (within-intervention group, ranging from 0.41 to 13.25). However, due to the heterogeneity between the studies, this review study may not be conclusive enough to make each intervention standardized. Further high-quality RCTs with larger sample sizes and more consistent methodologies are needed, and corrective findings from such studies should be applied to produce more robust evidence-based psychological interventions for treating alexithymia.

## 1. Introduction

The psychological term “alexithymia” was introduced by Sifneos in 1973 to describe specific characteristics observed in psychiatric patients having difficulties in identifying and expressing emotions. The patients are characterized by (1) difficulty in identifying and describing feelings; (2) difficulty in distinguishing between feelings and bodily sensations related to emotional activation; (3) restrained and limited imaginative processes, adopting the guise of an impoverished fantasy; and (4) an externally oriented cognitive style [1]. This clinically derived construct has been supported by a large accumulation of empirical research. There is a positive correlation with depression and anxiety [2], indicating that higher levels of alexithymia are associated with increased severity of these conditions such as autism, depressive disorders, anxiety disorders, schizophrenia, post-traumatic stress disorder (PTSD), and eating disorders. Moreover, chronic pain has strong links with alexithymia [3,4,5,6,7,8,9,10,11,12,13], which is prevalent in approximately 10% of the global population [14].

Not only correlated psychiatric disorders, alexithymia has also been found to be positively correlated with internal diseases such as diabetes, Parkinson’s disease, and coronary diseases [15,16,17,18,19].

Despite the widespread recognition of alexithymia as a concept, its underlying pathology remains a topic of debate. Attachment, adverse childhood experiences, and trauma are frequently cited as being related to alexithymia, with a substantial body of research exploring and supporting this association [20]. However, these findings do not fully uncover or explain the psychodynamic mechanisms underlying alexithymia.

The attention-appraisal model [21] explains how the mechanism of alexithymia impacts the ability to regulate emotions, based on how those emotions are appraised, but alexithymia impairs the capacity to appraise and regulate emotions, so people experience them in an undifferentiated manner. This model is based on findings from factor analysis and correlation analyses, but the accumulation of study has not reached to replace the original model of Sifneos [22]. Moreover, the attention-appraisal model has overlooked the role of imaginal activity as a component of the construct, which is also a critical factor in therapeutic interventions [23]. Lane and Schwartz’s cognitive-developmental theory [24] helps formulate a hypothesis on how the first step of “somatization” may occur. That is, the inability to perceive emotions through symbolic representations and signals of their own needs and desires, resulting in these internal states being instead referenced through somatic sensations or “somatization” [24]. Furthermore, the symptoms of alexithymia also encompass interpersonal elements. People high in alexithymia tend to make poorer decisions about selecting adaptive strategies, affecting quality of life by preventing connection with others as well as intimate relationships [25]. These actions hinder openness to therapists, potentially leading therapists to feel less effective and valued [26]. This challenge has been discussed as one of the contributing factors to the association between alexithymia and poor outcomes [27].

Although there is ongoing debate regarding the psychodynamic aspects of alexithymia, cognitive behavioral therapy (CBT) and its derivative psychotherapies have proved to be influential treatments of anxiety and depression [28,29], and the literature has demonstrated alexithymia mediates psychological etiology [30,31,32,33]; however, there has been a relative paucity of intervention studies on psychological treatments for improving alexithymia [34]. We found two systematic reviews on interventional studies of psychological therapy [35,36], but they are only thematized on dialectical behavioral therapy (DBT) and mindfulness-based interventions. The review on DBT [35] identified eight studies, of which six (75.0%) reported reductions in alexithymia after intervention. As for mindfulness-based interventions [36], four RCTs and a random-effects meta-analysis showed the effect of interventions to be significant to reduce TAS-20 (Toronto Alexithymia Scale) scores. While other psychological interventions for improving alexithymia are commonly used in clinical practice, no comprehensive review has been conducted to evaluate their efficacy across diverse methodologies and sample characteristics.

To address this gap, the present study aims to review focusing on randomized controlled trials (RCTs) investigating psychological treatments for alexithymia. Specifically, this review examines interventions excluding dialectical behavior therapy (DBT) and mindfulness-based interventions (MBIs), comparing them to treatment as usual or no treatment. The primary outcome measure across studies is the improvement in alexithymia symptoms, assessed using TAS-20. Using the PICOs framework, the populations considered include general populations and individuals with alexithymia. The interventions reviewed are psychological treatments excluding DBT and MBI, with comparisons to treatment as usual or no treatment. Outcomes are measured primarily using TAS-20.

## 2. Methods

### 2.1. Protocol and Registration

This systematic review was not registered with PROSPERO due to strict funding deadlines imposed by the grant agency. This expedited process left little opportunity for the formal registration of this systematic review in PROSPERO during the initial stages. Despite this, the review process adhered to PRISMA 2020 guidelines, ensuring methodological rigor, with emphasis on the search strategy, inclusion/exclusion criteria, and detailed documentation of the study selection process.

### 2.2. Search Strategy

We searched electronic databases (PubMed, PsycInfo, and Google Scholar) using the following terms: alexithymi*[ti] AND (intervention[ti] OR trial[ti] OR training[ti] OR treatment[ti] OR therapy[ti]) AND (2010:2024[pdat]), to investigate the literature published between 2010 and Oct 2024. We set this review’s starting point at 2010, since including studies from 2000 to 2010 could introduce variability in quality due to the advent of TAS-20 in 1994 [37]. At this point, no restrictions were applied, including duplication.

### 2.3. Eligibility

The following inclusion criteria were used: (1) published from 2010 to 2024; (2) full text available in English; (3) peer-reviewed journals; (4) RCT; and (5) baselines and outcomes measured by TAS-20 and raw data were provided. The final criteria are used because TAS-20 is the most widely used and reliable self-measurement [38]. This review was not limited to clinical/non-clinical situations or base diagnoses, age, or other participant demographics. We excluded non-psychological studies and also mindfulness and DBT since there were already systematic reviews on such therapies improving alexithymia [35,36]. Couple therapies and parent–child therapies were also excluded, as well as other therapies completed within less than one day.

### 2.4. Selection

This review is limited to RCTs with any control condition. The process of the selection is summarized in Figure 1. Initially, one reviewer assessed titles and abstracts based on predefined inclusion and exclusion criteria. In cases where eligibility was unclear from the title and abstract, the full text of the studies was carefully reviewed to make the decision. Eligible studies proceeded to a full-text review (N = 38). No additional reviewers were involved in the selection process. Removal of duplicates, screening, and data management were facilitated using Paperpile. A PRISMA flow diagram outlines the study selection process.

While only one reviewer was involved in the extraction process, a standardized data extraction form was developed to ensure consistency and to minimize potential bias or error. The extraction process was double-checked for accuracy by revisiting the original study details and verifying that all extracted data were consistent with the reported results.

Extracted information included study objectives, sample characteristics, size, inclusion/exclusion criteria, intervention details, assessment tools, statistical methods, results, and limitations.

## 3. Results

### 3.1. Summary of Included Studies

After screening, 18 studies with a combined total sample size of 1251 individuals met the criteria [39,40,41,42,43,44,45,46,47,48,49,50,51,52,53,54,55,56]; a flow diagram is illustrated in Figure 1, and a summary of the 18 studies is described in Table 1.

We found 15 studies that were 2-arm RCTs and 3 studies that were 3-arm RCTs, resulting in a total of 21 arms of interventions. Of these 21 arms, 14 arms (67%) investigated the effect of cognitive behavioral therapy on alexithymia, including acceptance and commitment therapy (7 arms), behavioral activation therapy (2 arms), cognitive behavioral therapy (CBT, 1 arm), schema therapy (1 arm), compassion-focused therapy (1 arm), group cognitive therapy (1 arm), and the Strength at Home Men’s Program (a cognitive-behavioral, trauma-informed group therapy program for active-duty or former military personnel who have engaged in recent physical intimate partner violence, 1 arm) [40,42,43,44,45,47,49,56]. There were seven arms of psychotherapies other than CBT, including interpersonal psychotherapy (IPT), meaning therapy, emotion-focused therapy, emotional intelligence training, emotion regulation training, multicomponent psychological intervention, and the mindtastic alexithymia app. Most of the studies were conducted in Iran (76.4%).

In all of the RCTs reviewed, the control groups consisted of participants who either did not receive any treatment or received treatment as usual.

### 3.2. Sample Characteristics

The sample size of each study ranged between 20 and 320. Among the 18 studies, samples of 10 studies were patients who had major diagnoses. There were six studies about physical diseases, including diabetes (three studies [42,43,51]), cancer (two studies [39,55]), and coronary heart diseases (one study [50]); three studies about mental disorders, including substance abusers (two studies [44,48]) and disruptive mood dysregulation disorder (one study [54]); and one study about psychosomatic disorders such as irritable bowel syndrome [53]. Five studies investigated women with relationship issues such as intimate partner violence, infertility, infidelity, and divorce [40,46,47,52,56]. There were three studies about volunteers, including high-alexithymia adults (TAS-20 ≥ 51) [41], female high school students [45], and nursing students [50].

### 3.3. Assessment of Alexithymia

Ten studies measured TAS-20 only on baseline and post-treatment, and seven studies [40,41,44,48,49,52,53] also investigated follow-up and reported further improvement over the follow-up period.

### 3.4. Pre- to Post-Treatment Effects

Seventeen studies reported significant reductions in TAS-20 in experimental groups, with further reductions in follow-up measurements (Table 2). Overall, compared with pre- to post-intervention, within-group effectiveness (Cohen’s d) ranged from 0.41 to 13.25 (medium to large).

Only one study reported a significant increase in TAS-20 scores (*p* = 0.001) [53].

### 3.5. Between-Group Effects

All 18 studies reported a significant reduction in alexithymia scores in the intervention groups compared to the control groups.

### 3.6. Risk of Bias Assessment

The revised Cochrane risk-of-bias tool for randomized trials (RoB 2) was used to assess the risk of bias in the final 18 studies included in this review. There are four types of risks: bias from the randomization process, bias due to deviations from intervention, bias due to missing outcome data, and bias in the measurement of the outcome. Although some studies in this review are described as quasi-experimental, they indicate that random assignment was employed, thus leading to their inclusion as RCTs. Among the final 18 studies selected for this systematic review, 4 studies explicitly mentioned the selection process [41,44,47,48], and 2 studies specify blinding was employed [44,48]. Bias in the measurement of the outcome may have existed in all studies since TAS-20 is self-reported.

## 4. Discussion

In addition to the previous two systematic reviews on mindfulness-based interventions and dialectical behavioral therapy (DBT) for improving alexithymia, this is the systematic review to summarize RCT studies on psychological treatments especially focused on alexithymia. Given the lack of sufficient intervention studies addressing alexithymia despite its clinical significance, and the fact that psychological therapies for this condition are already widely practiced in clinical settings, this study is intended to address the gap between the literature and clinical settings. We identified 18 studies that assessed the effectiveness of different psychotherapies, and all studies utilized TAS-20 as an outcome.

### 4.1. Clinical Implications

In 18 studies, 17 therapies demonstrated effectiveness in improving alexithymia, illustrating a showcase of treatments from each trial with different effect sizes (within-intervention group, ranging from 0.41 to 13.25). In this review, cognitive behavioral therapy was most common, indicated in 67% of the 18 studies. Not only CBT, but other therapies are also designed to increase awareness of interception and emotion and how to express them, which were already indicated in an earlier review [57]. The strategy to increase the capacity to direct attention toward emotional states and evaluate them accurately is recommended by both models of conceptualization of alexithymia: the original one by Sifneos [1] and the attention-appraisal model by Preece and Gross [58]. The original model indicates the importance of helping patients link their affects with images, utilize their imagination, and cultivate interests to discover their creative potential [59,60]. However, none of the 18 studies reviewed focused on therapies targeting these aspects. This highlights the need for further refinement of therapeutic approaches and additional research to address these critical components.

All three of the three-arm studies [43,47,56] compared ACT with other psychotherapeutic approaches: ACT versus BAT, ACT versus GCT, and ACT versus ST. Moreover, in each of these studies, the effect sizes reported for ACT exceeded those of the comparator therapies. This may be attributed to the distinctive model of ACT, which focuses on behavior change and enhancing psychological flexibility, grounded in relational frame theory (RFT) [60]. This might be the reason why ACT is well-suited for the treatment of alexithymia and somatization [61].

This review sought, as a supplementary objective, to offer clinicians evidence-based insights that may help refine therapeutic approaches and contribute to the development of personalized interventions for individuals with alexithymia. However, psychotherapies exhibited high variability in design and methodology, making integration difficult but offering the possibility for tailoring to variable diagnoses. It has been suggested that developing personalized psychological treatment should be based on individual analysis, not group analysis [62]. However, review of RCT is still needed to guide the future research for the development of treatments. This review helps identify effective therapeutic components and their mechanisms, match these components to patient subgroups based on factors like alexithymia severity or comorbid conditions, and adapt modular interventions to individual needs. More RCTs with active control groups would be helpful too.

This systematic review highlights a notable trend in the geographical distribution of research on alexithymia treatment, with a significant number of studies originating from Iran. One of the factors that may explain this predominance is that the Iranian Ministry of Health supports national mental health programs and research in accordance with the WHO’s Mental Health Gap Action Program, which prioritizes the treatment of disorders such as depression, anxiety, and bipolar disorder [63]. Secondly, the Farsi version of the Toronto Alexithymia Scale (TAS-20) has been validated and widely used in Iranian studies [64].

### 4.2. Limitations

The main methodological limitation of this current review is the heterogeneity of samples. The variety of psychotherapies and the inclusion of patients with major diseases and non-clinical individuals in the samples contributed to this heterogeneity, which is why meta-analysis was not performed. Another large limitation is the heterogeneity in interventions. Such variability makes it difficult to standardize each intervention. Furthermore, because TAS-20 is self-reported, the inherent bias is inevitable. It is known that highly alexithymic people may tend to reservedly assess their deficits [65]. In addition, TAS-20 is unable to assess the paucity of fantasies. To address these issues, clinician-rated measurements such as the Toronto Structured Interview for Alexithymia are suggested, though they have their own shortcomings in relation to cost and dependency on interviewer quality [66]. LEAS is another accessible psychological assessment that reflects participants’ conceptualization of alexithymia; however, its reliance on written responses introduces limitations, as it depends heavily on language ability and is susceptible to cultural biases [67]. The Canadian team of alexithymia researchers recommends using a multi-measure, multi-method approach for alexithymia measuring [68,69,70].

The third significant limitation of this systematic review is the geographic concentration of studies, with 76.4% originating from Iran. However, considering only the three RCTs conducted outside of Iran, all demonstrated effectiveness in improving alexithymia, which may apply to other regions. Furthermore, since many studies from Iran have focused on alexithymia among women in restrictive environments, the gender inequality present in Iran can be viewed as a microcosm of broader global disparities. Therefore, findings from these studies may have applicability to women across various cultural contexts worldwide.

The fourth limitation of this review is the potential presence of publication bias, as we did not perform specific tests, such as a funnel plot analysis or Egger’s test, in the assessment. This limitation could have influenced the overall conclusions drawn from the included studies.

The fifth limitation of this study is that this study is not pre-registered with PROSPERO. Future reviews should prioritize registration with PROSPERO prior to commencing the research process.

The sixth limitation of this study is that the selection process was conducted solely by a single author, which may have introduced potential bias or errors. It is recommended that future studies involve multiple reviewers to enhance reliability and reduce the risk of subjective bias.

As the seventh limitation, the literature search was limited to three databases: PubMed, PsycInfo, and Google Scholar. The inclusion of additional comprehensive databases in future reviews is encouraged to ensure a more exhaustive and representative selection of studies.

## 5. Conclusions

In this review, we identified 18 RCTs and 21 arms on psychotherapies aimed at improving alexithymia, and most reported significant decreases in TAS-20 scores among participants compared to control groups who either did not receive any treatment or received treatment as usual. These findings suggest that such psychological interventions might be effective for treating alexithymia. However, due to the heterogeneity of the studies, along with potential biases, the evidence remains inconclusive. Further high-quality RCTs with larger sample sizes and more consistent methodologies are needed to strengthen the evidence base and provide clearer guidance on effective treatments for alexithymia.

## Figures and Tables

**Figure 1 behavsci-14-01173-f001:**
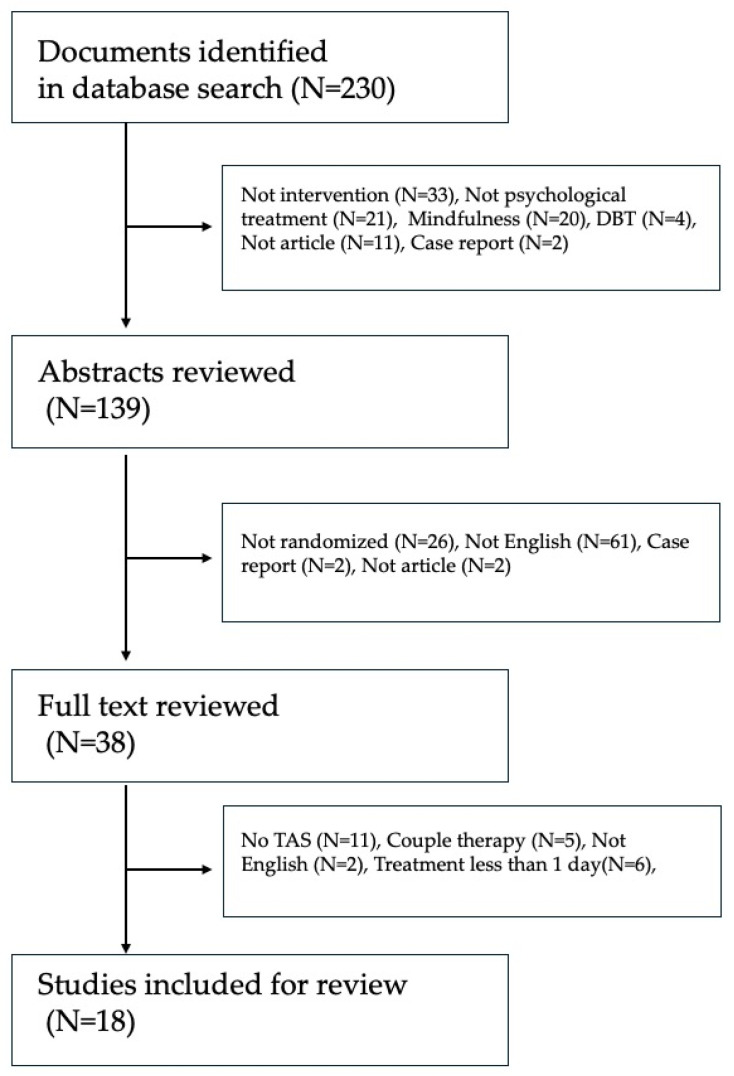
Flow diagram.

**Table 1 behavsci-14-01173-t001:** Study and sample characteristics.

Study	Sample	Inclusion or Exclusion Criteria	Therapy	TAS-20 Measuring	Results (Intervention vs. Control)
Porcelli et al. (2011) Italy [32]	N = 104 (55.8% female) M age = 47.3	Cancer patients, aged 18–70.	Multicomponent psychological intervention	Baseline, post-treatment (6 months)	Significantly improved TAS-20 (*p* < 0.001)
Berke et al. (2017) USA [33]	N = 135 (0% female) M age = 38.0	At least one act of male-to-female physical intimate partner violence (IPV) over the previous 6 months or severe physical IPV over the past 12 months, or an ongoing legal problem related to IPV.	A cognitive-behavioral, trauma-informed group therapy program for active-duty or former military personnel who have engaged in recent physical intimate partner violence (IPV): Strength at Home Men’s Program (SAH-M)	Baseline, post-treatment (3 month), 3-month follow-up	Significantly improved TAS-20 (*p* < 0.01)
Lukas et al. (2019) Germany [34]	N = 29 (53% female) M age = 26.8	TAS-20 ≥ 51, access to a smartphone (Android), sufficient German skill.	Mindtastic alexithymia app (MT-ALEX)	Baseline, post-treatment (2 weeks), 1-month follow-up	Significantly improved TAS-20 (*p* = 0.001)
Bahadori et al. (2021) Iran [35]	N = 28 (54% female) M age = 51.75	Patients with type 2 diabetes, age range of 30–70 years.	Compassion-focused therapy (CFT)	Baseline, post-treatment (4 weeks)	Significantly improved TAS-20 (*p* = 0.006)
NejadKazemfard et al. (2021) Iran [36]	N = 45	A minimum type 2 diabetes, duration of 6 months.	Behavioral activation therapy (BAT) or ACT	Baseline, post-treatment (8 weeks)	Significantly improved TAS-20 (both interventation groups) (*p* = 0.001). ACT more effective than BCT (*p* < 0.001)
Saeedi et al. (2021) Iran [37]	N = 20 M age = 38.0	Substance abuser according to the DSM5 criteria.	CBT	Baseline, post-treatment (10 weeks), 6 weeks follow-up	Significantly improved TAS-20 (both interventation groups) (*p* = 0.001)
Ghaznavi et al. (2022) Iran [38]	N = 118 (100% female) M age = 16.5	2nd-grade female high school students.	ACT	Baseline, post-treatment (8 weeks)	Significantly improved TAS-20 (*p* < 0.001)
Boorboor et al. (2022) Iran [39]	N = 118 (100% female)	Infertile women suffering from depression (based on the Beck Depression Inventory-II (BDI-II).	Behavioral activation therapy (BAT)	Baseline, post-treatment (6 weeks)	Significantly improved TAS-20 (*p* < 0.05)
Sadeghi et al. (2022) Iran [40]	N = 30 (100% female) M age = 27.5	Women who realized their spouse’s infidelity after at least two years of living together, age range of 25–45 years.	ACT or group cognitive therapy (GCT)	Baseline, post-treatment (12 weeks)	Significantly improved TAS-20 (both intervention groups) (*p* = 0.001). Significant difference between ACT and GCT (*p* = 0.002)
Saeedi et al. (2022) Iran [41]	N = 30 (0% female) M age = 38.3	Being a substance abuser according to DSM5 criteria.	Interpersonal psychotherapy (IPT)	Baseline, post-treatment (12 weeks), 6 weeks follow-up	Significantly improved TAS-20 (*p* < 0.05). Follow-up phase maintained significant improvement (*p* < 0.05).
Rahnama et al. (2022) Iran [42]	N = 24 M age = 39.1	With diagnosis of premature coronary heart disease and hypertension by a cardiologist based on the criteria of the World Health Organization (WHO) in patients to be confirmed. Being 30–50 years old.	ACT	Baseline, post-treatment (4 weeks), 3 months follow-up	Significantly improved TAS-20 (*p* < 0.0001). Follow-up phase maintained significant improvement (*p* < 0.0001).
Kamel et al. (2022) Saudi Arabia [43]	N = 70 (40.0% female)	Nursing students of Imam Abdulrahman Bin Faisal University.	Emotional intelligence training	Baseline, post-treatment (6 weeks)	Significantly improved TAS-20 (*p* = 0.005)
Naseri et al. (2023) Iran [44]	N = 320 (79.7% female) M age = 20.24	Patients aged 20 to 40 years with type 2 diabetes. Scored above the cutoff points of the questionnaires (Alexithymia Questionnaire, Body Image Concerns Questionnaire, Negative Automatic Thoughts Questionnaire).	ACT	Baseline, post-treatment (2 months)	Significantly improved TAS-20 (*p* < 0.001)
Karimi et al. (2023) Iran [45]	N = 30 (100% female)	Disclosure of their spouse’s infidelity for more than six months, no request for divorce.	ACT	Baseline, post-treatment (2.5 months), and follow-up	Significantly improved TAS-20 (*p* < 0.001)
Dana et al. (2023) Iran [46]	N = 30 (100% female) M age = 34.4	At least one year had passed since the diagnosis of irritable bowel syndrome (IBS) by a specialist.	Emotion-focused therapy	Baseline, post-treatment (12 weeks), 3 months follow-up	Significantly improved TAS-20 (*p* < 0.05)
Falah et al. (2023) Iran [47]	N = 30 (100% female)	Aged 13 to 16 years and have experience of disruptive mood dysregulation disorder.	Emotion regulation training	Baseline, post-treatment (2 months)	Significantly increased TAS-20 (*p* = 0.001)
Mahmoudi et al. (2024) Iran [48]	N = 30 (100% female) M age = 34.4	Women with breast cancer and being treated with cancer medical treatments.	Integrated approach of meaning therapy	Baseline, post-treatment (8 weeks)	Significantly improved TAS-20 (*p* = 0.02)
Almardani etal. (2024) Iran [49]	N = 60 (100% female) M age = 30.0	Divorced and referring to a welfare organization, no remarriage, age range of 19 to 35 years, obtaining a high score on the alexithymia scale.	ACT or schema therapy (ST)	Baseline, post-treatment (15 weeks)	Significantly improved TAS-20 (both intervention groups) (*p* = 0.002). Effectiveness of ACT significantly higher than ST (*p* = 0.002)

M age, mean age; TAS-20, Toronto Alexithymia Scale; ACT, acceptance and commitment therapy; CBT, cognitive behavioral therapy.

**Table 2 behavsci-14-01173-t002:** Study outcomes at post-treatment and follow-up on TAS-20.

	M(SD) for the Intervention Group at Pre-Treatment	M(SD) for the Intervention Group at Post-Treatment	Difference in Scores (Pre-Score Minus Post-Score)	Pre–Post: Within-Intervention Group Effect Sizes (Cohen’s d)	M(SD) for the Intervention Group at Follow-Up	Pre-F/U: Within-Intervention Group Effect Sizes (Cohen’s d)	M(SD) for the Control Group at Pre-Treatment	M(SD) for the Control Group After Treatment	Pre–Post: Within-Control Group Effect Sizes (Cohen’s d)	M(SD) for the Control Group at Follow-Up	Pre-F/U: Within-Control Group Effect Sizes (Cohen’s d)
Porcelli 2011 [32]	50.90 (13.36)	43.17 (12.32)	−7.73	0.60			51.29 (15.03)	60.46 (14.80)	−0.61		
Berke 2017 [33]	56.58 (15.26)	53.01 (13.36)	−3.57		51.7 (13.79)		57.91 (10.99)	58.48 (11.99)		57.99 (12.74)	
Lukas 2019 [34]	63.60 (6.70)	59.73 (6.62)	−3.87	0.58	57.07 (8.63)	0.84	58.50 (7.65)	54.63 (7.08)	0.53	59.83 (7.47)	−0.18
Bahadori 2021 [35]	62.40 (6.46)	53.66 (8.67)	−8.74	1.14			57.23 (8.35)	58.76 (9.38)	−0.17		
NejadKazemfard 2021 [36]	BAT 78.93 (3.84) ACT 79.20 (4.17)	37.80 (6.62) 26.23 (4.56)	−41.13 −52.97	7.60 12.12			80.26 (3.12)	78.10 (2.85)	0.73		
Saeedi 2021 [37]	41.15 (7.92)	37.50 (6.50)	−3.65	0.50	38.72 (6.42)	0.34	40.30 (7.00)	39.24 (6.52)	0.16	40.64 (7.19)	−0.05
Ghaznavi 2022 [38]	63.10 (6.06)	36.05 (6.32)	−27.05	4.36			63.25 (7.81)	64.30 (4.24)	−0.16		
Boorboor 2022 [39]	34.73 (12.78)	29.07 (14.67)	−5.66	0.41			27.53 (10.37)	32.80 (12.36)	−0.46		
Sadeghi 2022 [40]	ACT 67.0 (4.37) GCT 67.7 (9.1)	58.60 (7.13) 55.0 (8.3)	−8.4 −12.7	1.42 1.46			68.9 (7.7)	67.31 (5.6)	0.24		
Saeedi 2022 [41]	42.82 (7.60)	36.42 (8.40)	−6.4	0.80	37.55 (7.25)	0.71	40.30 (7.00)	39.24 (6.52)	0.16	40.64 (7.19)	−0.05
Rahnama 2022 [42]	67.48 (1.54)	39.80 (2.52)	−27.68	13.25	46.14 (4.15)	6.82	69.66 (5.90)	65.40 (8.48)	0.58	60.91 (5.70)	1.51
Kamel 2022 [43]	58.86 (9.02)	47.51 (12.84)	−11.35	1.02			57.80 (10.03)	53.83 (9.50)	0.41		
Naseri 2023 [44]	65.20 (4.48)	46.27 (2.96)	−18.93	4.99			66.60 (7.53)	67.20 (7.27)	−0.08		
Karimi 2023 [45]	68.06 (14.20)	55.26 (6.72)	−12.8	1.15	54.26 (5.88)	1.27	66.66 (10.81)	66.80 (8.38)	−0.01	67.60 (4.96)	−0.11
Dana 2023 [46]	66.20 (9.94)	44.82 (6.52)	−21.38	2.54			59.08 (2.88)	59.67 (3.58)	−0.81		
Falah 2023 [47]	54.68 (6.41	71.73 (6.94)	17.05	−2.56			55.24 (6.64)	56.28 (5.35)	−0.17		
Mahmoudi 2024 [48]	49.86 (10.16)	41.98 (10.38)	−7.88	0.77			48.33 (10.68)	47.68 (10.94)	0.06		
Almardani (2024) [49]	ACT 64.35 (3.11) ST 63.90 (2.32)	40.10 (1.27) 44.60 (1.74)	−24.25 −19.3	10.21 9.41			54.25 (1.73)	53.03 (1.96)	0.66		

BAT, behavioral activation therapy; ACT, acceptance and commitment therapy; GCT, group cognitive therapy; ST, schema therapy.

## Data Availability

The template data collection forms, data extracted from included studies, data used for all analyses, analytic code, and any other materials utilized in this review are available upon request. Interested parties may contact the correspondent author for access to these materials.

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
