# Peer review of "Psychological Treatments for Alexithymia: A Systematic Review"

_behavsci, 2024, doi:10.3390/bs14121173_

Round 1

Reviewer 1 Report

Comments and Suggestions for Authors

The abstract offers a very comprehensive suumary on psychological interventions for treating alexithymia, it's current prevelance and limitations in current research in treating it.  

Although not necessary, explaining why the study is not registered with PROSPERO would be helpful as it will help strengten the methodological approach of the paper. A brief expansion on the methodology mentioning adherence to PRISMA guidlines, maybe a brief description of the inlcusion and exclusion criterai in the abstract would be valuable. 

Given majority of the studies were conducted in Iran, are there any cultural, social, or healthcare sysyemic impacts on the applicability to other regions? 

Highlighting limitations to the TAS-20 as a self reported tool, leading to inherent bias, and if there are any alternatives/supplementa to the TAS-20 for future studies could help improve accuracy of measurement. 

Author Response

Dear reviewer,

Thank you for giving us the opportunity to submit a revised draft of our manuscript titled “Psychological treatments for alexithymia: a systematic review” to Behavioral Sciences. We appreciate the time and effort that you and the reviewers have dedicated to providing your valuable feedback on our manuscript. We are grateful to the reviewers for their insightful comments on our paper.

We have revised the manuscript accordingly, and the changes are highlighted using a track change function. We have also copied the changes to this response letter, where they are underlined.

Here is a point-by-point response to the reviewers’ comments and concerns.

Although not necessary, explaining why the study is not registered with PROSPERO would be helpful as it will help strengthen the methodological approach of the paper. A brief expansion on the methodology mentioning adherence to PRISMA guidlines, maybe a brief description of the inlcusion and exclusion criteria in the abstract would be valuable. 

→Thank you for your comment. This systematic review was not registered with PROSPERO due to timing constraints and limited resources.

We added the following sentences to the Methods section.

“This systematic review was not registered with PROSPERO due to strict funding deadlines imposed by the grant agency. This expedited process left little opportunity for the formal registration of the systematic review in PROSPERO during the initial stages. Despite this, the review process adhered to PRISMA 2020 guidelines, ensuring methodological rigor, with emphasis on the search strategy, inclusion/exclusion criteria and detailed documentation of the study selection process.”

We also added the following sentences to the Limitations section.

“The fifth limitation is not pre-registered with PROSPERO. Future reviews should prioritize registration with PROSPERO prior to commencing the research process.”

→Regarding PRISMA, the review process adhered to PRISMA 2020 guidelines and the corresponding diagram is provided as Appendix A. According to your suggestion, we added the following sentences to the Abstract section.

“The objectives of this systematic review were to explore and synthesize findings from recent randomized controlled trials (RCTs) about psychological treatments, with inclusion criteria; 1) published from 2010 to 2024; 2) full text available in English; 3) peer reviewed journal; and 4) baselines and outcomes were measured by TAS-20 and raw data was provided, excluding non-psychological studies and Mindfulness and DBT therapy.”

Given majority of the studies were conducted in Iran, are there any cultural, social, or healthcare sysyemic impacts on the applicability to other regions? 

→Thank you for your comment. We believe there are no significant impacts with the applicability, as even when considering only the three RCTs conducted outside of Iran, all demonstrated effectiveness in improving alexithymia. We added this point to the limitations.

“The third significant limitation of this systematic review is the geographic concentration of studies, with 76.4% originating from Iran. However, considering only three RCTs were conducted outside of Iran, all demonstrated effectiveness in improving alexithymia, which may apply to other regions. Furthermore, since many studies from Iran have focused on alexithymia among women in restrictive environments, and the gender inequality present in Iran can be viewed as a microcosm of broader global disparities. Therefore, findings from these studies may have applicability to women across various cultural contexts worldwide.”

Highlighting limitations to the TAS-20 as a self-reported tool, leading to inherent bias, and if there are any alternatives/supplementa to the TAS-20 for future studies could help improve accuracy of measurement. 

→ Thank you for your comment. The inherent bias of TAS-20 being a self-reported measure, and an alternative way (clinician-rated measurements) was suggested in 3.6. Risk of bias assessment and the limitation.

We added the following sentences about Levels of Emotional Awareness scale (LEAS) as another alternative of TAS-20 as to the limitations.

“LEAS is another, accessible psychological assessment which reflects participants' conceptualization of alexithymia; however, its reliance on written responses introduces limitations, as it depends heavily on language ability and is susceptible to cultural biases. [63]”

Reviewer 2 Report

Comments and Suggestions for Authors

The paper addresses alexithymia, which is undoubtedly an interesting and timely topic. Below are some suggestions that could enhance the comprehensiveness of the work:

  1. In the introduction, on line 40, the disorder should be referred to as post-traumatic stress disorder (PTSD).

  2. The introduction could also include an explanation of the psychodynamic basis of alexithymia, given that alexithymia is primarily a concept rooted in the psychodynamic perspective. Additionally, it would be beneficial to provide an overview of the psychodynamic background of alexithymia in relation to somatic, or rather, somatoform and psychosomatic disorders. This is one of the strengths of the paper, as it connects a psychodynamic phenomenon with contemporary psychotherapeutic techniques such as DBT and CBT.

  3. In the methodology section, it would be helpful to clarify why the studies included in the systematic review span the past 14 years, rather than, for example, the last 10 or 20 years.

  4. Since the majority of studies originate from a single country, this could be further elaborated upon in the discussion.

  5. Additionally, the discussion should be expanded to address the association of alexithymia with psychosomatic illnesses, specifically somatoform and psychosomatic diagnoses (pain syndromes and autoimmune diseases), which could also be further examined in the discussion.

Overall, the paper addresses a fascinating and contemporary subject, and with these revisions, it could prove valuable to the professional community.

Author Response

Dear reviewer,

Thank you for giving us the opportunity to submit a revised draft of our manuscript titled “Psychological treatments for alexithymia: a systematic review” to Behavioral Sciences. We appreciate the time and effort that you and the reviewers have dedicated to providing your valuable feedback on our manuscript. We are grateful to the reviewers for their insightful comments on our paper.

We have revised the manuscript accordingly, and the changes are highlighted using a track change function. We have also copied the changes to this response letter, where they are underlined.

Here is a point-by-point response to the reviewers’ comments and concerns.

1. In the introduction, on line 40, the disorder should be referred to as post-traumatic stress disorder (PTSD).

→Thank you for pointing out. I corrected it as post-traumatic stress disorder (PTSD).

2. The introduction could also include an explanation of the psychodynamic basis of alexithymia, given that alexithymia is primarily a concept rooted in the psychodynamic perspective. Additionally, it would be beneficial to provide an overview of the psychodynamic background of alexithymia in relation to somatic, or rather, somatoform and psychosomatic disorders. This is one of the strengths of the paper, as it connects a psychodynamic phenomenon with contemporary psychotherapeutic techniques such as DBT and CBT.

→Thank you for your comment. We added an explanation of the psychodynamic background to enhance the persuasiveness towards the objective. Changed sentences are as follows in the introduction.

“Not only correlated psychiatric disorders, alexithymia has also been found to be positively correlated with internal diseases such as diabetes, Parkinson’s disease and coronary diseases [15-19]. The psychodynamic basis of alexithymia is explained by the attention-appraisal model [20], in which the possible mechanism alexithymia serves as a foundation for various mental disorders. Within this theory, people decide how to regulate emotions based on how it is appraised, but alexithymia impairs that capacity, experiencing their emotions in an undifferentiated manner. From the viewpoint of the psychodynamic background, this also explains how the first step of “somatization” may occur. Inability to perceive emotions through symbolic representations and signals of their own needs and desires, resulting in these internal states are instead referenced through somatic sensations or “somatization” [21]. Furthermore, people high in alexithymia tend to make poorer decisions about selecting adaptive strategies, affecting quality of life by preventing connection with others as well as intimate relationships [22]. These actions hinder openness to therapists, potentially leading therapists to feel less effective and valued [23]. This challenge has been discussed as one of the contributing factors to the association between alexithymia and poor outcomes [24].”

3. In the methodology section, it would be helpful to clarify why the studies included in the systematic review span the past 14 years, rather than, for example, the last 10 or 20 years.

→Research on treatments for alexithymia began to accelerate in the late 2000s, particularly after the advent of Toronto Alexithymia Scale (TAS-20) in 1994. Including studies until 2010 could introduce variability in quality. Thus, we set the review's starting point in 2010 to ensure a more consistent methodological approach. We added following sentences to 2.2. Search strategy.

”We set the review's starting point at 2010, since including studies from 2000-2010 could introduce variability in quality, due to the advent of TAS-20 in 1994. [34]. ”

4. Since the majority of studies originate from a single country, this could be further elaborated upon in the discussion.

→Thank you for your comment. We believe there are no issues with the applicability, as even when considering only the three RCTs conducted outside of Iran, all demonstrated effectiveness in improving alexithymia. We added this point to the limitations.

“The third significant limitation of this systematic review is the geographic concentration of studies, with 76.4% originating from Iran. However, considering only the three RCTs conducted outside of Iran, all demonstrated effectiveness in improving alexithymia, which may apply to other regions. Furthermore, since many studies from Iran have focused on alexithymia among women in restrictive environments, and the gender inequality present in Iran can be viewed as a microcosm of broader global disparities. Therefore, findings from these studies may have applicability to women across various cultural contexts worldwide.”

5. Additionally, the discussion should be expanded to address the association of alexithymia with psychosomatic illnesses, specifically somatoform and psychosomatic diagnoses (pain syndromes and autoimmune diseases), which could also be further examined in the discussion.

→Thank you for this pointing out. We ended up addressing the the association of alexithymia with psychosomatic illnesses in mostly introduction for the better structure and focusing more on results in discussion. Here are sentences added to the discussion according to your suggestion.

“All three of the three-arm studies compared ACT with other psychotherapeutic approaches; ACT versus BAT, ACT versus GCT, and ACT versus ST. Moreover, in each of these studies, the effect sizes reported for ACT exceeded those of the comparator therapies. This may be attributed for the distinctive model of ACT, which focuses on behavior change and enhancing psychological flexibility, grounded in Relational Frame Theory (RFT)[56]. This might be the reason why ACT is well-suited for the treatment of alexithymia and somatization [57]”

Reviewer 3 Report

Comments and Suggestions for Authors

The manuscript provides a comprehensive review of psychological treatments for alexithymia, a condition often linked to emotional suppression, depression, and anxiety. Despite its prevalence among psychiatric patients, therapeutic interventions for alexithymia have not been extensively studied. The authors focus on recent randomized controlled trials (RCTs) between 2010 and 2024, excluding previously reviewed mindfulness-based and dialectical behavioral therapies (DBT), to evaluate other approaches like cognitive behavioral therapy (CBT) and its variations, including acceptance and commitment therapy, behavioral activation therapy, schema therapy, and compassion-focused therapy. The review assesses treatment outcomes using the Toronto Alexithymia Scale (TAS-20) across 18 RCTs, which include 21 treatment arms and involve a total of 1,251 participants. The manuscript identifies promising results, with most interventions showing statistically significant reductions in TAS-20 scores, indicating improved alexithymia symptoms. However, a major issue highlighted is the heterogeneity among the studies, with effect sizes ranging from 0.41 to 13.25. This variability limits the ability to draw standardized conclusions across treatments. The authors suggest several critical improvements for future studies, including the need for larger sample sizes and more consistent methodologies. These improvements would help in creating more reliable, evidence-based psychological interventions that could better address alexithymia in clinical practice. The manuscript underscores the potential of CBT-based treatments but emphasizes that without methodological rigor and consistency, it is challenging to establish a standardized approach for treating alexithymia. This review provides  valuable insights and highlights essential areas for refinement, setting the stage for future research that could yield more definitive, actionable recommendations for clinicians.

Author Response

Dear reviewer,

Thank you for giving us the opportunity to submit a revised draft of our manuscript titled “Psychological treatments for alexithymia: a systematic review” to Behavioral Sciences. We appreciate the time and effort that you and the reviewers have dedicated to providing your valuable feedback on our manuscript.

We are grateful for your kind feedback and for highlighting the strengths of our work, as well as the areas where further research is needed.

As there were no specific changes or revisions requested, we are glad to hear that the manuscript meets your expectations. Please let us know if there are any additional aspects you would like us to address or clarify further.

Thank you again for your time and effort in reviewing our work. We look forward to the next steps in the publication process.

Reviewer 4 Report

Comments and Suggestions for Authors

Dear Authors, 

Thank you for your interesting paper “Psychological treatments for alexithymia: a systematic review”. You will find hereunder some comments: 

Abstract:

Lines 8- 9: The authors do not explicitly state whether the correlation between alexithymia and conditions like depression and anxiety is positive or negative (the same observation applies to the lines 37-38). Please elaborate.  Moreover, including a statement about how the risk of bias was evaluated would strengthen the abstract.

Introduction:

Line 33: the expression “psychological condition term” sounds odd. Moreover, the expression “to describe model psychiatry patients" in the same line is unclear. Please reconsider.

The authors mention a long list psychological and physical conditions to which alexithymia is linked. The context is missing. Providing an explanation on why alexithymia is related both domains, and creating a "bridge” between them domains, would improve the reader's experience.

The introduction would benefit from a stronger transition into the study's objectives; in the current version, objective of the study is mentioned at the end of the introduction.

Methods:

PROSPERO registration is a significant element of systematic review. The "timing constraints” mentioned by the authors seem vague; more specific explanation would be beneficial. Given given the absence of PROSPERO registration, for transparency reasons, it would be beneficial to specify which elements of PRISMA were particularly emphasized.

The search was limited to PubMed, PsycInfo, and Google Scholar. Are there any particular reasons of selecting these three data bases?

The date range (2010–2024) is clearly stated, which is good. Please clarify why 2010 was chosen as the starting point.

Please provide further details related to the screening process. How the articles have been assessed for eligibility? How many researchers have been involved in the process?

How were duplicates later handled?

Only one reviewer has been involved in the data extraction process, which could introduce bias or error. Could you elaborate on how you can ensure that none of this actually happened?

The PICOs framework is not clearly defined.

Line 84-85: Exclusion criteria: “Couple therapies, parent-child therapies and therapies completing within less than one day were also excluded”. Does it mean that the intervention that lasted more than one day were included?

Results: 

The fact that 76.4% of the studies were conducted in Iran is interesting. The authors could, however, elaborate on whether this regional concentration might affect generalizability.

Some of the studies assessed alexithymia at follow-up. Can the authors specify the the follow-up time points?

Discussion:

Line 202: "Most psychotherapies employed... demonstrated effectiveness" - the sentence is vague, please provide examples. 

The last parts of the paper – Discussion, and conclusions, the paper felt not specific enough. Elaborating the discussion about the implications of the findings, addressing limitations in a deeper and more precise way, and providing concrete suggestions for future research, would strengthen the narration.

Overall, the specificity of treatment alexithymia in different psychotherapies should be clearly stated. See papers by Bagby et al., Taylor et al., Preece et al.

Author Response

Dear reviewer,

Thank you for giving us the opportunity to submit a revised draft of our manuscript titled “Psychological treatments for alexithymia: a systematic review” to Behavioral Sciences. We appreciate the time and effort that you and the reviewers have dedicated to providing your valuable feedback on our manuscript. We are grateful to the reviewers for their insightful comments on our paper.

We have revised the manuscript accordingly, and the changes are highlighted using a track change function. We have also copied the changes to this response letter, where they are underlined.

Here is a point-by-point response to the reviewers’ comments and concerns.

Abstract:

Lines 8- 9: The authors do not explicitly state whether the correlation between alexithymia and conditions like depression and anxiety is positive or negative (the same observation applies to the lines 37-38). Please elaborate.  Moreover, including a statement about how the risk of bias was evaluated would strengthen the abstract.

→Thank you for pointing these out. Here is the sentences we added to Abstract according to your suggestion.

“Alexithymia, a psychological condition characterized by emotional suppression, is positively correlated with depression and anxiety, and can develop into various mental disorders.”

We also added the following sentences to Introduction section.

“This positive correlation with depression and anxiety [2], indicates that higher levels of alexithymia are associated with increased severity of these conditions such as autism, depressive disorders, anxiety disorders, schizophrenia, post-traumatic stress disorder (PTSD) and eating disorder. ”

Introduction:

Line 33: the expression “psychological condition term” sounds odd. Moreover, the expression “to describe model psychiatry patients" in the same line is unclear. Please reconsider.

→Thank you for noticing this for us. We changed Introduction as following sentences.

“The psychological term "alexithymia" was introduced by Sifneos in 1973 to describe specific characteristics observed in psychiatric patients having difficulties in identifying and expressing emotions. The patients are characterized by: 1) difficulty in identifying and describing feelings; 2) difficulty in distinguishing between feelings and bodily sensations related to emotional activation; 3) restrained and limited imaginative processes, adopting the guise of an impoverished fantasy; and 4) externally orientated cognitive style [1].”

The authors mention a long list psychological and physical conditions to which alexithymia is linked. The context is missing. Providing an explanation on why alexithymia is related both domains, and creating a "bridge” between them domains, would improve the reader's experience.

The introduction would benefit from a stronger transition into the study's objectives; in the current version, objective of the study is mentioned at the end of the introduction.

→Thank you for your comment. We added the explanation of the mechanisms that link various disorders and improved the whole introduction to clarify the flow to strengthen the transition into the study’s objectives. Following sentences are added to the introduction.

”The psychodynamic basis of alexithymia is explained by the attention-appraisal model [20], in which the possible mechanism of alexithymia serves as a foundation for various mental disorders. Within this theory, people decide how to regulate emotions  based on how it is appraised, but alexithymia impairs that capacity, experiencing their emotions in an undifferentiated manner. From the viewpoint of the psychodynamic background, this also explains how the first step of those somatization may occur. Inability to perceive emotions through symbolic representations and signals of their own needs and desires, resulting in these internal states instead referenced through somatic sensations or somatization [21]. Furthermore, people high in alexithymia tend to make poorer decisions about selecting adaptive strategies, affecting quality of life by preventing connection with others as well as intimate relationships [22]. These actions hinder openness to therapists, potentially leading therapists to feel less effective and valued [23]. This challenge has been discussed as one of the contributing factors to the association between alexithymia and poor outcomes [24].

Cognitive behavioral therapy (CBT) and its derivative psychotherapies have proved to be an influential treatment of anxiety and depression [25,26], and literature  has demonstrated alexithymia mediates psychological etiology [27-30]; however there has been a relative paucity of intervention studies on psychological treatments for improving alexithymia [31]. We found 2 systematic reviews on interventional studies of psychological therapy [32,33], but they are only thematized on Dialectical Behavioral Therapy (DBT) and mindfulness-based interventions. The review on DBT [32] identified 8 studies, of which 6 (75.0%) reported reductions in alexithymia after intervention. As for mindfulness-based interventions, four RCTs and a random-effects meta-analysis showed the effect of interventions to be significant to reduce TAS-20 (Toronto Alexithymia Scale) scores. While other psychological interventions for improving alexithymia are commonly used in clinical practice, no comprehensive review has been conducted to evaluate their efficacy across diverse methodologies and sample characteristics.

To address this gap, the present study aims to review focusing on randomized controlled trials (RCTs) investigating psychological treatments for alexithymia. Specifically, the review examines interventions excluding Dialectical Behavior Therapy (DBT) and Mindfulness-Based Interventions (MBI), comparing them to treatment as usual or no treatment. The primary outcome measure across studies is the improvement in alexithymia symptoms, assessed using TAS-20. Using the PICOs framework, the populations considered include general populations and individuals with alexithymia. The interventions reviewed are psychological treatments excluding DBT and MBI, with comparisons to treatment as usual or no treatment. Outcomes are measured primarily using the TAS-20.”

Methods:

PROSPERO registration is a significant element of systematic review. The "timing constraints” mentioned by the authors seem vague; more specific explanation would be beneficial. Given given the absence of PROSPERO registration, for transparency reasons, it would be beneficial to specify which elements of PRISMA were particularly emphasized.

→Thank you for your comment. This systematic review was not registered with PROSPERO due to timing constraints and limited resources.

We added the following sentences to the Methods section.

“This systematic review was not registered with PROSPERO due to strict funding deadlines imposed by the grant agency. This expedited process left little opportunity for the formal registration of the systematic review in PROSPERO during the initial stages. Despite this, the review process adhered to PRISMA 2020 guidelines, ensuring methodological rigor, with emphasis on the search strategy, inclusion/exclusion criteria and detailed documentation of the study selection process.”

We also added the following sentences to the Limitations section.

“The fifth limitation is not pre-registered with PROSPERO. Future reviews should prioritize registration with PROSPERO prior to commencing the research process.”

The search was limited to PubMed, PsycInfo, and Google Scholar. Are there any particular reasons of selecting these three data bases?

→Thank you for questioning. We selected PubMed, PsycInfo, and Google Scholar because they provide comprehensive coverage of biomedical, psychological, and interdisciplinary research. Other databases such as Scopus, Web of Science, or Embase were not included because they can occur overlap with those data bases as they index many of the same journals especially in the fields of psychology, psychiatry, and related biomedical research. Even so we added this point to the limitation.

”As the seventh limitation, the literature search was limited to three databases: PubMed, PsycInfo, and Google Scholar. The inclusion of additional comprehensive databases in future reviews is encouraged to ensure a more exhaustive and representative selection of studies.”

The date range (2010–2024) is clearly stated, which is good. Please clarify why 2010 was chosen as the starting point.

→Research on treatments for alexithymia began to accelerate in the late 2000s, particularly after the advent of Toronto Alexithymia Scale (TAS-20) in 1994. Including studies until 2010 could introduce variability in quality. Thus, we set the review's starting point at 2010 to ensure a more consistent methodological approach. We added following sentences to 2.2. Search strategy.

”We set the review's starting point at 2010, since including studies from 2000-2010 could introduce variability in quality, due to the advent of TAS-20 in 1994. [34]. ”

Please provide further details related to the screening process. How the articles have been assessed for eligibility? How many researchers have been involved in the process? How were duplicates later handled?

Only one reviewer has been involved in the data extraction process, which could introduce bias or error. Could you elaborate on how you can ensure that none of this actually happened?

→Thank you for your comment. We added more information on how the articles assessed with these sentences to methods section.

“Initially, one reviewer assessed titles and abstracts based on predefined inclusion and exclusion criteria. In cases where eligibility was unclear from the title and abstract, the full text of the studies was carefully reviewed to make the decision. Eligible studies proceeded to a full-text review(N=38). No additional reviewers were involved in the selection process. Removal of duplicates, screening and data management were facilitated using Paperpile. A PRISMA flow diagram outlines the study selection process.

While only one reviewer was involved in the extraction process, a standardized data extraction form was developed to ensure consistency, and to minimize potential bias or error. The extraction process was double-checked for accuracy by revisiting the original study details and verifying that all extracted data was consistent with the reported results. “

We also added the following sentences to the Limitations section.

“The sixth limitation of this study is that the selection process was conducted solely by a single author, which may have introduced potential bias or errors. It is recommended that future studies involve multiple reviewers to enhance reliability and reduce the risk of subjective bias.”

The PICOs framework is not clearly defined.

→Thank you for your comment. According to your suggestion, we added this sentences to the end of the introduction.

“Specifically, the review examines interventions excluding Dialectical Behavior Therapy (DBT) and Mindfulness-Based Interventions (MBI), comparing them to treatment as usual or no treatment. The primary outcome measure across studies is the improvement in alexithymia symptoms, assessed using TAS-20. Using the PICOs framework, the populations considered include general populations and individuals with alexithymia. The interventions reviewed are psychological treatments excluding DBT and MBI, with comparisons to treatment as usual or no treatment. Outcomes are measured primarily using the TAS-20.”

Line 84-85: Exclusion criteria: “Couple therapies, parent-child therapies and therapies completing within less than one day were also excluded”. Does it mean that the intervention that lasted more than one day were included?

→Thank you for your comment. Please let me clarify. Only psychotherapies involving a series of sessions, rather than one-time interventions, were included. We changed the sentences of the end of 2.3. Eligibility as follows.

“Couple therapies and parent-child therapies were also excluded, as well as other therapies completed within less than one day.”

Results: 

The fact that 76.4% of the studies were conducted in Iran is interesting. The authors could, however, elaborate on whether this regional concentration might affect generalizability.

→Thank you for your comment. We believe there are no issues with the applicability, as even when considering only the three RCTs conducted outside of Iran, all demonstrated effectiveness in improving alexithymia. We added this point to the limitations.

“The third significant limitation of this systematic review is the geographic concentration of studies, with 76.4% originating from Iran. However, considering only three RCTs were conducted outside of Iran, all demonstrated effectiveness in improving alexithymia, which may apply to other regions. Furthermore, since many studies from Iran have focused on alexithymia among women in restrictive environments, and the gender inequality present in Iran can be viewed as a microcosm of broader global disparities. Therefore, findings from these studies may have applicability to women across various cultural contexts worldwide.”

Some of the studies assessed alexithymia at follow-up. Can the authors specify the the follow-up time points?

→Thank you for your comment. The follow-up period is specified in the 'TAS-20 Measuring' column of Table 1.

Discussion:

Line 202: "Most psychotherapies employed... demonstrated effectiveness" - the sentence is vague, please provide examples. 

The last parts of the paper – Discussion, and conclusions, the paper felt not specific enough. Elaborating the discussion about the implications of the findings, addressing limitations in a deeper and more precise way, and providing concrete suggestions for future research, would strengthen the narration. Overall, the specificity of treatment alexithymia in different psychotherapies should be clearly stated. See papers by Bagby et al., Taylor et al., Preece et al.

→Thank you for your insightful comments. I have revised the entire discussion section and deepened its content. Renewed discussion section is as follows.

“4. Discussion

In addition to the previous 2 systematic reviews on mindfulness-based interventions and dialectical behavioral therapy (DBT) for improving alexithymia, this is the systematic review to summarize RCT studies on psychological treatments especially focused on alexithymia. Given the lack of sufficient intervention studies addressing alexithymia despite its clinical significance, and the fact that psychological therapies for this condition are already widely practiced in clinical settings, this study is intended to address the gap between literature and clinical settings. We identified 18 studies that assessed the effectiveness of different psychotherapies, and all studies utilized TAS-20 as an outcome.

4.1. Clinical implications

In 18 studies, 17 therapies demonstrated effectiveness in improving alexithymia, illustrating a showcase of treatments from each trial with different effect sizes (within-intervention group, ranging from 0.41 to 13.25). In this review, cognitive behavioral therapy was most common, indicated in 67% of the 18 studies. Not only CBT, but other therapies are also designed to increase awareness of interception and emotion and how to express them, which were already indicated in an earlier review [54]. The effectiveness of these interventions may support and further substantiate the existing theoretical framework; the attention-appraisal model. This emphasizes that the capacity to direct attention toward emotional states and evaluate them accurately is a critical limiting factor in the effectiveness of emotion regulation strategies [55]. It is reasonable to consider that these psychotherapiesmay improve alexithymia by enhancing emotion regulation and expression skills.

All three of the three-arm studies compared ACT with other psychotherapeutic approaches; ACT versus BAT, ACT versus GCT, and ACT versus ST. Moreover, in each of these studies, the effect sizes reported for ACT exceeded those of the comparator therapies. This may be attributed for the distinctive model of ACT, which focuses on behavior change and enhancing psychological flexibility, grounded in Relational Frame Theory (RFT)[56]. This might be the reason why ACT is well-suited for the treatment of alexithymia and somatization [57]

This review sought, as a supplementary objective, to offer clinicians evidence-based insights that may help refine therapeutic approaches and contribute to the development of personalized interventions for individuals with alexithymia. However, psychotherapies exhibited high variability in design and methodology, making integration difficult but offering the possibility for tailoring to variable diagnoses. It has been suggested that developing personalized psychological treatment should be based on individual analysis, not group analysis [58] However, review of RCT is still needed to guide the future research for development of treatments. This review helps identifying effective therapeutic components and their mechanisms, matching these components to patient subgroups based on factors like alexithymia severity or comorbid conditions, and adapting modular interventions to individual needs. More RCTs with active control groups would be helpful too.

This systematic review highlights a notable trend in the geographical distribution of research on alexithymia treatment, with a significant number of studies originating from Iran. One of the factors that may explain this predominance is that the Iranian Ministry of Health supports national mental health programs and research, in accordance with WHO's Mental Health Gap Action Program, which prioritizes the treatment of disorders such as depression, anxiety, and bipolar disorder [59]. Secondly, the Farsi version of the Toronto Alexithymia Scale (TAS-20) has been validated and widely used in Iranian studies [60].

4.2. Limitations

The main methodological limitation of this current review is the heterogeneity of samples. The variety of psychotherapies, and the inclusion of patients with major disease and non-clinical individuals in the samples, contributed to this heterogeneity, which is why meta-analysis was not performed. Another large limitation is the heterogeneity in interventions. Such variability makes it difficult to standardize each intervention. Furthermore, because TAS-20 is self-reported, the inherit bias is inevitable. Also, highly alexithymic people may tend to reservedly assess their deficits [61]. To address this issue, clinician-rated measurements such as the Toronto Structured Interview for Alexithymia are suggested, though they have their own shortcomings in relation to cost and dependency on interviewer quality [62]. LEAS is another, accessible psychological assessment which reflects participants' conceptualization of alexithymia; however, its reliance on written responses introduces limitations, as it depends heavily on language ability and is susceptible to cultural biases. [63]

The third significant limitation of this systematic review is the geographic concentration of studies, with 76.4% originating from Iran. However, considering only the three RCTs conducted outside of Iran, all demonstrated effectiveness in improving alexithymia, which may apply to other regions. Furthermore, since many studies from Iran have focused on alexithymia among women in restrictive environments, and the gender inequality present in Iran can be viewed as a microcosm of broader global disparities. Therefore, findings from these studies may have applicability to women across various cultural contexts worldwide.
       The fourth limitation of this review is the potential presence of publication bias, as we did not perform specific tests, such as a funnel plot analysis or Egger’s test, in the assessment. This limitation could have influenced the overall conclusions drawn from the included studies.

The fifth limitation of this study is this study is not pre-registered with PROSPERO. Future reviews should prioritize registration with PROSPERO prior to commencing the research process.

The sixth limitation of this study is that the selection process was conducted solely by a single author, which may have introduced potential bias or errors. It is recommended that future studies involve multiple reviewers to enhance reliability and reduce the risk of subjective bias.

As the seventh limitation, the literature search was limited to three databases: PubMed, PsycInfo, and Google Scholar. The inclusion of additional comprehensive databases in future reviews is encouraged to ensure a more exhaustive and representative selection of studies.”

Round 2

Reviewer 4 Report

Comments and Suggestions for Authors

Several statements are misleading. For instance, "The psychodynamic basis of alexithymia is explained by the attention-appraisal model [20], in which the possible mechanism alexithymia serves as a...". The attention-appraisal model of alexithymia does not describe the psychodynamic basis.

See for review: Taylor, G. J., Porcelli, P., Bagby, R. M. (2024). Alexithymia: a defense of the original conceptualization of the construct and a critique of the attention-appraisal model. Clinical Neuropsychiatry, 21(5), 329-357.

Please make sure that all theoretical foundations are clearly explained.

In the current form, theoretical background of this paper is not sufficient. Please see the above-described paper to describe and highlight differences in treatment of alexithymia based on two different models of alexithymia. There is a strong need to refer to these models when critically analysing the studies in this systematic review.

Different models/conceptualizations as well as treatments methods has their specificity in treatment of alexithymia. This should be highlighted and discussed.

Author Response

Dear Reviewer,

Thank you for giving us the opportunity to submit a revised draft of our manuscript titled “Psychological Treatments for Alexithymia: A Systematic Review” to Behavioral Sciences. We sincerely appreciate the time and effort that you and the reviewers have dedicated to providing such thoughtful and constructive feedback on our manuscript.

Your insightful comments have been immensely helpful in improving the quality of our paper and have enhanced our understanding of alexithymia. We are truly grateful for your contributions, which have not only strengthened the manuscript but also deepened our perspective on this topic.

We have revised the manuscript in accordance with your suggestions, and the changes are highlighted using the track change function. Additionally, we have included the corresponding revisions in this response letter, where they are underlined for clarity.

We sincerely hope that our revisions meet your expectations and address all concerns, and we look forward to your further feedback.

Round 3

Reviewer 4 Report

Comments and Suggestions for Authors

Thank you for your work, the paper is ready to be published.